# Psychotherapeutic Interventions to Improve Psychological Adjustment in Type 1 Diabetes: A Systematic Review

**DOI:** 10.3390/ijerph182010940

**Published:** 2021-10-18

**Authors:** Davinia M. Resurrección, Desirée Navas-Campaña, Mencía R. Gutiérrez-Colosía, Joaquín A. Ibáñez-Alfonso, Desireé Ruiz-Aranda

**Affiliations:** 1Department of Psychology, Universidad Loyola Andalucía, 41704 Seville, Spain; dmresurreccion@uloyola.es (D.M.R.); dmnavas@uloyola.es (D.N.-C.); jaibanez@uloyola.es (J.A.I.-A.); daruiz@uloyola.es (D.R.-A.); 2Human Neuroscience Lab, Universidad Loyola Andalucía, 41704 Seville, Spain

**Keywords:** diabetes type 1, psychological intervention, psychological adjustment, emotional intervention, systematic review, mental health

## Abstract

*Background*: International clinical practice guidelines highlight the importance of improving the psychological and mental health care of patients with Type 1 diabetes mellitus (T1DM). Psychological interventions can promote adherence to the demands of diabetes self-care, promoting high quality of life and wellbeing. *Methods*: A systematic review was carried out to determine whether psychological treatments with a specific focus on emotional management have an impact on glycemic control and variables related to psychological adjustment. Comprehensive literature searches of PubMed Medline, Psycinfo, Cochrane Database, Web of Science, and Open Grey Repository databases were conducted, from inception to November 2019 and were last updated in December 2020. Finally, eight articles met inclusion criteria. *Results*: Results showed that the management of emotions was effective in improving the psychological adjustment of patients with T1DM when carried out by psychologists. However, the evidence regarding the improvement of glycemic control was not entirely clear. When comparing adolescent and adult populations, findings yielded slightly better results in adolescents. *Conclusions*: More rigorous studies are needed to establish what emotional interventions might increase glycemic control in this population.

## 1. Introduction

Type 1 diabetes mellitus (T1DM) is a leading chronic disease associated with significant mortality and economic cost worldwide. Currently, there is a growing prevalence of this disease in both the child–youth population and in adults. According to a recent review and metanalysis, the incidence between 1990 and 2019 was 15 per 100,000 inhabitants and the prevalence was 9.5% in the world [1,2,3].

The American Diabetes Association recommends a glycated hemoglobin (HbA1c) target value below 7.5% for all age groups. Keeping glucose levels within the indicated target delays diabetic complications [4]. For the management and control of diabetes, patients are required to adhere to a complex daily therapeutic plan that involves self-monitoring of glucose, insulin administration, a diet plan, and regular physical exercise among others [1]. Uncontrolled T1DM is associated with physiological complications like hypoglycemia, hyperglycemia, and ketoacidosis. These complications, if recurrent, can lead to serious medical problems including neurological damage, kidney disease, vision loss, and vascular damage, as well as reduced quality of life [5]. Despite the risks generated by poor management, more than 70% of people with T1DM maintain HbA1c levels above 7%, and less that 20% of them are able to achieve optimal control over blood glucose [5,6]. One of the main reasons that patients experience above-target glycemic control is the challenge of managing diabetes [7]. This is partly attributable to psychological factors that adversely affect self-management.

Difficulty controlling diabetes causes psychological problems such as depression, emotional burdens, stress, and worries that result from dealing with a demanding chronic disease and are highly prevalent in this population. This is known as diabetes distress (DD) and has been associated with difficulties in glycemic control and poor self-care behavior [8]. The presence of this symptomatology has been associated with decreases in the quality-of-life levels [9,10,11] and lower general wellbeing [12].

Assessment of psychological and social status should be included as part of a comprehensive approach to people with diabetes. Systematizing the assessment of the psychological state will help mitigate the psychological impact of diabetes [13]. In fact, clinical practice guidelines [14] highlight the importance of improving care for the psychological and mental health problems of patients with T1DM.

Appropriate knowledge and information about the disease do not necessarily translate into a better metabolic control. This may be due to the fact that lifestyle changes necessary for metabolic control entail working on other aspects that influence decision making such as management of unpleasant emotions [15]. Some studies emphasize the association between attachment orientation and adherence to glucose monitoring. Attachment theory supports that a secure relationship with the caregiver in early life will lead to normal emotional development and better adaptation to diabetes [16]. Therefore, the ability of the individual to process and manage emotional states can affect the relationship between negative emotions experienced and diabetes management [17].

Reviews and metanalyses have shown contradictory results on the influence of psychological interventions in improving glycemic control. Two metanalyses of studies carried out with adolescents and adults with diabetes, reported having achieved a significant improvement in HbA1c compared to poor control at baseline [18,19]. However, the metanalysis carried out by Winkley et al. [20] found no overall effect of psychological interventions compared to control groups on HbA1c. The discrepancy between results could be explained as a consequence of the limited number of studies identified [20] or because the other metanalyses were either focused on mindfulness-based interventions [19] or included only an adolescent population [18]. In this sense, there is research that supports the concept that the intensity of psychological interventions (determined by the number and duration of sessions) and motivational interviews are related to a reduction in DD and HbA1c. In general, when the study focuses on assessing the efficacy of interventions to produce beneficial behavioral changes for the management of diabetes, psychological interventions obtain positive results [19,21,22]. So far, reviews have analyzed the impact of psychological interventions on diabetes management, ignoring the effect that the components of such interventions might have [20]. These previous systematic reviews have some limitations: (a) they have not specifically focused on emotional components included in the psychological intervention; (b) they have included interventions carried out by a generalist interventionist; (c) they have included a population with psychological symptoms such as subclinical depression; (d) they have included a population with different types of diabetes.

Therefore, the main goal of this study was to systematically review the available literature on psychological treatments with a specific focus on the management of an emotional component and their impact on both glycemic control and psychological adjustment. A secondary objective was the evaluation of the differences found in adults and children–adolescents, regarding the impact of the intervention on metabolic control.

## 2. Materials and Methods

PRISMA guidelines for reporting systematic reviews were followed [23] and the protocol was registered in PROSPERO on 28 April 2020 (registration No.: CRD42020159017) with the last update on 23 February 2021. Comprehensive literature searches of PubMed Medline, Psycinfo, Cochrane Database, Web of Science, and Open Grey Repository databases were conducted, from inception to November 2019 and were last updated in December 2020. Databases were searched separately by two reviewers (D.M.R. and D.N.). The search strategy incorporated combinations of three different concepts: (a) interventions; (b) type 1 diabetes; and (c) psychological wellbeing. The search was piloted in PubMed and then adapted to run across the other databases (see Appendix A). For the identification of additional articles, reference lists of the included studies as well as recent reviews in the field were checked. In addition, expert authors in the field were contacted.

### 2.1. Elegibility Criteria

#### 2.1.1. Selection of Studies

First, duplicated studies were deleted. Second, title and abstract study selection was done in duplicate (D.R.A. and D.N.). Third, based on the screening of title and abstract a selection of potentially relevant articles was made in duplicate (D.R.M. and M.R.) and a third reviewer participated in cases of disagreement (D.R.A.). Finally, after reading the full text, a final selection was made. The Kappa inter-agreement statistic was good (κ: 0.5; 95% CI: 0.303–0.797).

Selected studies met specific inclusion criteria (see Table 1). The review was focused exclusively on selecting those studies that performed a psychological intervention with a main emotional component. To analyze the effect of interventions with an emotional component, prospective cohort studies with a comparison group or randomized controlled studies were included. The allowed comparators were other active treatment, usual care, or waiting list. The outcome variables required in these studies were the analysis of any psychological adjustment variable and glycemic control. The target population for analysis was children–adolescents and/or adults with diagnosis of type 1 diabetes. Because a diagnosis of diabetes is frequently followed by a period of remission, usually partial [24], patients must have been diagnosed for at least 12 months. Patients who had not received insulin treatment in the last year, or who had very severe medical conditions, were not included.

#### 2.1.2. Data Extraction

The data extraction sheet was pilot tested and refined accordingly. The main characteristics of these studies were rigorously extracted by M.R. and verified by a second reviewer (D.R.A.). For each study, information was collected about the author(s), year of publication, study country, sample size, mean age, mean diabetes duration, study conditions, intervention duration, provider and format, follow-up assessments, and main results.

### 2.2. Risk of Bias in Individual Studies

Quality assessment was performed independently in duplicate (D.M.R. and D.N.) and a third reviewer participated in case of disagreement (D.R.A.). The level of agreement was excellent (intraclass correlation coefficient, 0.81; 95% CI, 0.72–0.89). The quality for RCT studies was assessed with the RoB 2 tool [25]. This tool is the second version of the Cochrane tool for assessing risk of bias in randomized trials. The RoB 2 tool assesses five ‘risk of bias’ criteria: randomization process, deviations from intended interventions, missing outcome data, measurement of the outcome, and selection of the reported result. The risk of bias judgments for each domain are: low risk, some concerns, or high risk. This tool makes an algorithm that maps responses to signaling questions to a proposed risk of bias judgment for each domain. The quality of prospective cohort studies was assessed with the Newcastle–Ottawa Scale (NOS) [26]. The NOS awards stars for eight items, clustered into three categories: selection of study groups, comparability of the groups, and the ascertainment of either the exposure or outcome for cohort studies.

The present review did not require the approval of an ethics committee.

## 3. Results

### 3.1. Search Results

The search strategy produced 3545 potentially relevant studies (see Figure 1 PRISMA flow diagram). One article was identified from the references of the articles selected. Of these, 666 were removed as duplicates. Of the remaining studies, 2800 were excluded after reviewing the title and abstract. Of the 79 articles selected for full-text reviewing, 72 were excluded for the following main reasons: 24 because the provider was not a psychology professional; 21 because they were either protocol studies or were not an RCT or a prospective cohort study; 11 did not fulfil the criteria of at least 12 months of T1D diagnosis; 6 did not include a psychological treatment, did not have an emotional component, or did not have a control group; 3 included a population with other psychological diagnoses; and 7 were not possible to find. Finally, eight studies were selected.

### 3.2. Study Quality

The results of the quality assessment of the included RCT are presented in Figure 2 and Appendix A. According to qualitative criteria by the RoB 2 tool [25], there was no study that entirely showed a ‘low risk’. Two RCTs had ‘some concerns’ about a risk of bias and five RCTs had a ‘high risk’ of bias. The level of risk of bias per domain can be seen in Appendix A. NOS assessment [26] regarding risk of bias in the cohort study showed a ‘high risk’: in the selection category it obtained less than 3 stars (high risk), in the comparability category it obtained less than 2 stars (high risk), in the outcome category it obtained two stars (low risk) (see Appendix A).

### 3.3. Synthesis of Results

Eight independent studies met the predefined inclusion criteria. A full overview of the studies is presented in Table 2. Of these, seven were randomized controlled trials and one was a non-randomized trial in which participants that postponed their participation in the study were used as controls [27]. The comparator in three studies was routine diabetes care [28,29,30], another three used alternative interventions like support visits [31], the Dutch adaptation of blood glucose awareness training (BGAT) [32], a diabetes education program (KnowIt) [33], and two studies used waiting-lists [27,34].

Regarding location, two studies were carried out in the United States [29,33], four studies were conducted in Europe [27,28,32,34], one in the United Kingdom [31], and one in Australia [30]. The eight studies comprised 935 participants, 374 men and 561 women, mean age ranging from 13.1 to 47.3 years. Studies that focused on children and/or adolescent populations had a range of diabetes duration of 5.1 to 9.2 years [29,30,31] whereas in studies focused on adults or on all populations, the range varied from 16.6 to 26.1 years [27,28,32,33,34].

### 3.4. Intervention

Cognitive-behavioral therapy (CBT) was the base intervention used by four trials to improve different psychological variables including emotions related to diabetes [28,30,32,34]. The intervention ‘StyrKRAFT i Ditt Liv’ (power to choose your direction) applied a structured manual of techniques that included relaxation and a logbook for self-care activities and feelings of stress [28]. The ‘Dia-fit’ intervention for chronic fatigue was composed of eight modules, one of them specific to reducing diabetes-related distress [34]. The ‘Best of Coping’ (BOC) intervention for psychological adjustment used a combination of techniques for coping, conflict resolution, and cognitive restructuring distributed in ten modules [30]. Finally, the CBT group-based intervention, in comparison to BGAT for self-efficacy and diabetes distress, applied goal-setting and cognitive restructuring techniques on different themes like the role of cognition and emotions in diabetes self-care and the stress [32]. Additionally, one trial based its diabetes-related stress intervention on multisystemic therapy, that targeted adherence problems within the family system, the peer network, and the broader community, using a set of techniques based on CBT, parent training, and behavioral family systems [29]. Another study applied motivational interviewing to improve psychological functioning [31]. One trial was based on the comparison of an education intervention (Knowlt) for the management of diabetes and an emotion regulation program (Ontrack) based on motivational interviewing, labelling feelings, and separating feelings from appraisals of self-worth among other techniques for the development of personalized emotion management [33]. Additionally, there was one study that used a specific psychological support program for quality of life, psychological adjustment, and self-efficacy [27]. Instruments like role-playing, metaplan, and problem-solving were used in the sessions to address, for example, the management of health loss and dysfunctional thoughts including emotions.

The intervention was led by a psychologist (health and clinical) in six studies [27,29,30,31,33,34] and by a multidisciplinary team including a diabetes specialist nurse and a psychologist in two studies [28,32]. When describing the therapy process carried out in the studies, the following can be noted: in four studies, delivery consisted of weekly two-hour sessions over a range of five to eight weeks for groups of five to twelve participants [27,28,30,32]; one study included two or three home-based sessions per week until goal achievement [29]; another study consisted of participant-determined appointments for 20–60 min over a year [31]; in another study the delivery was a one-day group workshop and four one-hour online video meetings over three months [33]; finally, a study included five to eight face-to-face sessions of 50 min and eight web-based modules with a global duration of five months [34]. In all the studies, the follow-up period ranged from 6 to 12 months.

### 3.5. Outcomes

Three of eight studies included an adolescent population [29,30,31]. Four of the eight interventions proved beneficial for metabolic control with reductions in HbA1c concentrations [27,28,31,33], only one of them focusing on adolescents [31]. Related to psychological adjustment, all the included studies reported improvements. Five studies found improvements in DD [28,29,32,33,34], and three studies found less anxiety and depressive symptoms [27,30,31]. Regarding type of intervention, CBT was beneficial for decreasing fatigue severity and functional impairment [34], diabetes stress and depression [32], and developing better psychosocial wellbeing [28,30]; multisystemic therapy showed a significant reduction in diabetes stress [29]; motivational interviewing was effective for enhancing psychological adjustment and quality of life [31]; the emotion-focused approach led to reductions in DD [33]; the psychological support improved DD, health-related quality of life, and self-efficacy [27].

## 4. Discussion

The main aim of the present review was to synthesize the evidence about psychological treatments with a specific focus on the management of emotional support or other psychological variables that could have an impact on glycemic control and wellbeing, in order to clarify the state of the art in the field and suggest future directions. To the best of our knowledge, this is the first systematic review specifically examining interventions delivered by a psychology professional. This is a relevant aspect because, as has been seen, the improvement of stress management skills is an important factor for the proper management of the disease, adherence to treatment, and general wellbeing. Attachment theory supports the fact that patients internalize early experiences with caregivers that will determine whether others can be trusted; this influences their participation in the patient–provider relationship and ultimately, adherence to treatment [35]. But, whether therapists (non-psychology professionals) have competencies to deliver a specific psychological training is not clear, and has been highlighted as a limitation by a recent review [22]. Recent studies on chronic diseases indicate that physicians, nurses and other health professionals often lack the training and skills that psychologists have, to deliver behavior-based treatments, and also highlight that interprofessional teams yield the most positive outcomes [36,37,38]. However, there are few randomized controlled studies in which appropriately qualified psychologists carry out these interventions.

Taken together, the studies included in this review suggested that psychological interventions focused on emotional components were more effective than control conditions in improving psychological adjustment in adolescents and adults with T1DM. The results of the studies included in the present review showed that the ability to use emotional information may be important for patients with T1DM to better manage their disease. Patients with better emotional strategies can handle the negative emotions that are associated with their condition, buffering the impact the disease has on them [17]. In this line, having poor emotional management could be an impediment to the health for patients with diabetes. A possible explanation could be the relationship between experiencing negative emotionality and the onset of DD. When a patient finally experiences DD, they are more likely to reduce their self-care [39,40,41]. The improvements in psychological functioning showed the impact that interventions on emotional management have on DD in patients with diabetes. It extends the evidence and confirms the beneficial impact psychosocial interventions have in patients with T1DM. Furthermore, given the efficacy of the interventions it would be important to include them in routine care.

Regarding HbA1c levels, four of the eight interventions proved beneficial for metabolic control with reductions in HbA1c concentrations [27,28,31,33]. Although the evidence is weak on the effectiveness of psychological treatments to improve HbA1c, it seems that the studies that found metabolic improvements contain elements that may be relevant for the design of future interventions. First, all studies except one were applied both in group and individual formats, which allowed advantages of group processes to be combined with elements of necessary individualization. In this sense, the studies that did not find reductions in HbA1c were delivered either only in group format or individually, or in family format. Second, self-care behaviors were central and explicit targets in the interventions, which can help maintain an adequate HbA1c level. Third, some of the studies have patients with elevated HbA1c at baseline which may respond better to the intervention. Psychological interventions with an emotional component might generate an improvement in HbA1c levels by reducing DD and stress levels [39,40]. However, more studies are needed to explore this relationship.

A secondary aim of this review was the assessment of possible differences between adults and children–adolescents with T1DM in psychosocial and metabolic outcomes after receiving a specific psychological treatment. Three of the reviewed studies focused on adolescents [29,30,31] showed improvements in the diabetes-related distress, as well as improvements in other measured factors like wellbeing, psychological adjustment, quality of life, and self-efficacy [30,31]. Moreover, two of the three psychological interventions focused on adolescents also showed improvements in metabolic control [29,31]. The third study only found a marginal effect (*p* = 0.058) over glycemic control at 12 months follow-up assessment [30]. These results are similar to the studies that included an adult population [28,32,33]. Even though studies with adolescents included in the present systematic review are fewer than those carried out with adults, it can be pointed out that the impact of interventions with emotional management could be more relevant in adolescents due to the difficulties of glycemic control at this stage of life. However, one thing in common is the effect of the interventions over time; in both populations, the long-term effects (12 months) are more visible and stable than in the short-term (3 months). These results are in line with those found in previous metanalyses [18,19].

The findings of this study should be interpreted in light of some limitations. First, the inclusion criteria for this review were limited to randomized controlled trials and cohort studies, as these kind of studies usually provide stronger evidence in the literature. Thus, it is possible that quasi-experimental studies may be underrepresented, losing complementary evidence about the effectiveness of psychological treatments over T1DM glycemic control and wellbeing. The small number of studies makes it difficult to draw conclusions on potentially differential effects of the interventions. Second, this study has evaluated the effect of psychological interventions with an emotional component delivered by psychologists but its comparative effect with respect to other interventions or health professionals has not been evaluated. Third, diverse primary outcomes make interpretation about effective psychological interventions more difficult. This small number of studies and the heterogeneity of the primary outcomes made the development of concurrent metanalysis infeasible. In order to compare the effects of psychological interventions with an emotional management component in different populations, it would have been desirable that: (a) the number of studies of both adolescent and adult populations included in the review were balanced; (b) there was sufficient research where the effect of the interventions would have been statistically weighted in both samples; (c) an appropriate number of studies of both samples could have been obtained to be able to carry out a metanalysis.

Despite these limitations, our study has several strengths. This is the first systematic review that includes interventions carried out specifically by psychology professionals. Future studies in this regard may be oriented to explore the psychologist–patient relationship within the framework of attachment theory. Second, only studies carried out with patients with T1DM after the ‘honeymoon period’ were included. This is relevant because during the first 12 months after the diagnosis, there are several factors that can affect the metabolic control of diabetes. Finally, focusing on the emotional component of psychological interventions allows future interventions to explore how it relates to improved self-management in diabetes.

## 5. Conclusions

Psychological interventions with an emotional component have been demonstrated to be effective in improving psychological adjustment in patients with T1DM when these interventions are carried out by a psychologist. Nevertheless, this psychological improvement was not always reflected in significantly better HbA1c levels. The studies reviewed included multicomponent and different therapeutic interventions. Further studies are needed to confirm what causes the change in these patients. Following the recommendations of the international diabetes guidelines [42] and taking into account the results of the studies analyzed in this review, there is a need for developing individualized interventions, integrating evidence-based programs in regular healthcare, and creating psychological resources to adjust to health challenges in diabetes.

## Figures and Tables

**Figure 1 ijerph-18-10940-f001:**
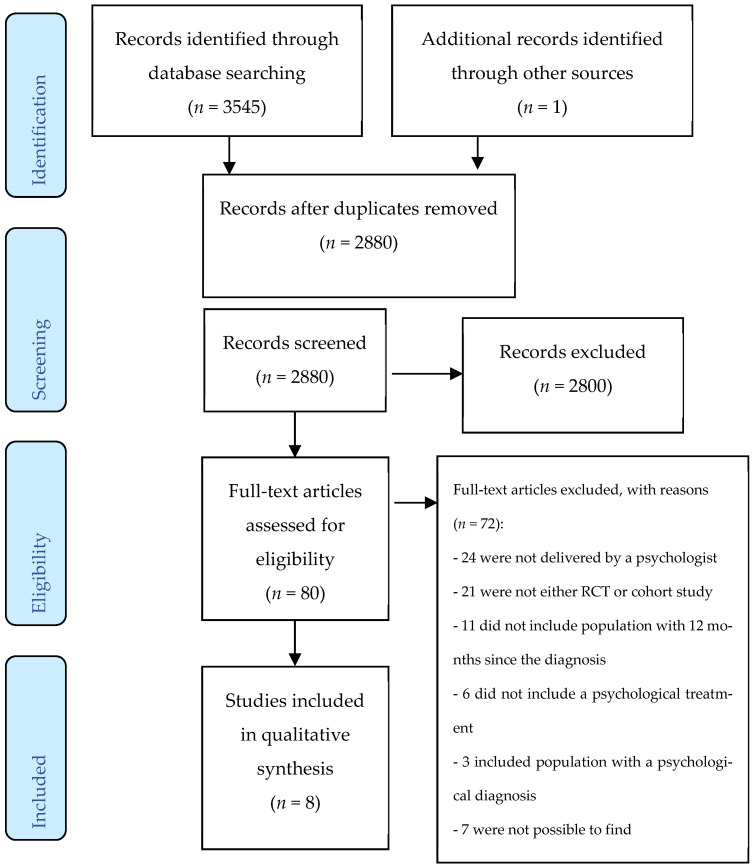
PRISMA flow diagram.

**Figure 2 ijerph-18-10940-f002:**
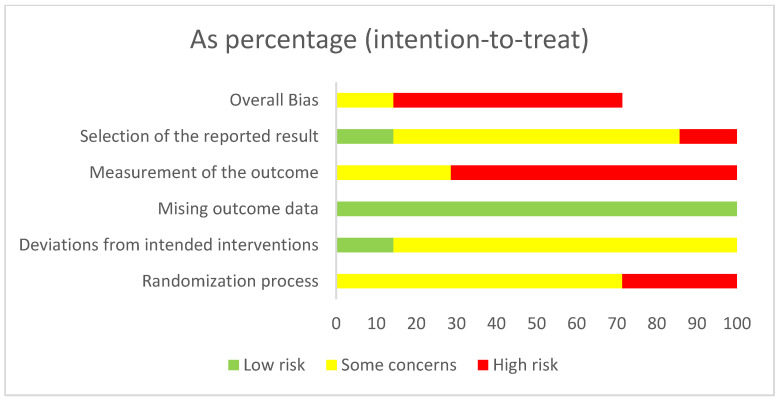
Quality assessment using RoB 2 Tool.

**Table 1 ijerph-18-10940-t001:** Inclusion and exclusion criteria for the studies included in the review.

Aspects Considered	Inclusion Criteria	Exclusion Criteria
Population	Adolescents and adults with type 1 diabetes mellitus with at least one year since diagnosis and insulin treatment	Presenting any severe medical or psychological condition, less than a year of diabetes diagnosis or not having an insulin treatment.
Outcome	Quality of life and glycemic control	The study only assesses metabolic variables or other variables. The independent effect of the intervention could not be determined (e.g., combining medication and psychological intervention).
Design	Randomized controlled trials, prospective cohorts with control group.	Qualitative studies, systematic reviews and/or meta-analysis, protocols, clinical cases, and editors’ letters, cross-sectional studies, retrospective cohorts.
Type of intervention	Psychological interventions with any emotional component, delivered in group or individual format. Interventions delivered by a psychologist accompanied or not by another specialist	Interventions delivered by any specialist without a psychologist.
Language	All languages	None
Setting	All settings are included	None
Comparator	Waiting list, usual care, or any active control	No comparator.

**Table 2 ijerph-18-10940-t002:** Characteristics of studies included in the review.

First Author, (Year), Country	Sample (Control/Intervention)	Years Mean (SD)	Mean (SD) Diabetes Duration, Years	Study Conditions: 1. Intervention 2. Control	Intervention Duration; Format; Provider	Follow-Up Assessment	Primary Outcome Metabolic; Psychological
Ellis et al. [29], USA	127 (63/64)	Adolescent population I: 13.4 (1.9) C: 13.1 (2.0)	I: 5.3 (3.9) C: 5.2 (4.8)	1. Multisystemic therapy 2. Standard care	6 months, 2–3 times per week; Home-based family sessions; Therapist	Month 7	Month 7 - Improved outcomes: Diabetes-related distress - Not improved outcomes: Metabolic control
Van der Ven et al. [32], The Netherlands	88 (45/43)	Adult population 37.8 (10.6)	18.0 (10.4)	1. Cognitive Behavior Therapy (CBT) based group training (CBGT) 2. Dutch adaptation of blood glucose awareness training (BGAT)	6 weekly 2 h sessions; group sessions (6–8p); diabetes nurse and psychologist	Month 3	Month 3 - Improved outcomes: Diabetes-distress (PAID) Self-efficacy (CIDS) Depressive symptoms (CES-D) - Not improved outcomes: Metabolic control HbA1c levels
Channon et al. [31], UK	66 (38/28)	Adolescent population I: 15.3 (0.97) C: 15.4 (1.19)	I: 9.2 (1.96) C: 9.1 (1.47)	1. Motivational interviewing 2. Support visits	12 months; individual frequency, home-based; health psychologist trainee	Months 12 and 24	Month 12 - Improved outcomes: Metabolic control HbA1c levels Quality of life (DQoLY)—satisfaction, impact, worries subscales Wellbeing (WBK)—depression, anxiety, positive wellbeing subscales Personal models of illness (PMDQ) Month 24 - Improved outcomes: Metabolic control HbA1c levels Quality of life (DQoLY)—satisfaction, impact subscales Wellbeing (WBK)—anxiety subscale Personal models of illness (PMDQ)—life worry subscale
Amsberg et al. [28], Sweden	74 (36/38)	Adult population I: 41.1 (11.7) C: 41.4 (12.9)	I: 19.9 (9.4) C: 23.2 (11.8)	1. CBT based intervention 2. Continuous glucose monitoring system	8 weekly 2 h sessions; group (4–6p) and individual; diabetes nurse and psychologist	Months 6 and 12	Month 6 - Improved outcomes: Glycemic control Diabetes distress (PAID) Month 12 - Improved outcomes: Glycemic control Diabetes distress (PAID) Wellbeing (W-BQ12) Anxiety (HAD) Depression (HAD) Perceived Stress (PSS)
* Forlani et al. [27], Italy	55 (33/22)	Adult population I: 40.7 (12.0) C: 39.4 (12.7)	I: 16.6 (11.5) C: 16.6 (8.7)	1. Psychological support program 2. Waiting list	7 weekly 2 h sessions; groups (8–12p); psychologist	Month 6	Month 6 - Improved outcomes: Metabolic control HbA1c levels Depression (BDI) Anxiety(SAS) Wellbeing (WED)
Serlachius et al. [30], Australia	104 (30/74)	Adolescent population I: 14.6 (1.16) C: 14.3 (1.12)	I: 5.97 (3.12) C: 6.12 (3.80)	1. CBT based program 2. Standard care	5 weekly 2 h sessions; group sessions; health psychologist	Months 3 and 12	Month 3 - Improved outcomes: Diabetes distress (DSQ) Self-efficacy (SED) Quality of life (DQoL) Month 12 - Improved outcomes: Self-efficacy (SED) Quality of life (DQoL) - Not improved outcome: Glycemic control
Menting et al. [34], The Netherlands	120 (60/60)	Adult population I: 44.4 (12.1) C: 42.9 (12.5)	I: 24.2 (13.3) C: 24.1 (13.9)	1. CBT 2. Waiting list	5 months; 5–8 individual face-to-face sessions + web-based modules; clinical psychologists	Month 6	Month 6 - Improved outcomes: Fatigue severity (CIS) Functional impairment
Fisher et al. [33], USA	301 (Knowlt 149/ Ontrack 152)	Adult population Knowlt: 47.3 (14.5) Ontrack: 42.8 (15.1)	Knowlt: 26.1 (14.0) Ontrack: 23.2 (13.3)	1. Improving emotion regulation skills (OnTrack) 2.Education/behavior change intervention (Knowlt)	3 months; 1-day group + 4 online videos; psychologist (OnTrack) diabetes nurse (Knowlt)	Month 9	Month 9 - Improved outcomes: Metabolic control HbA1c levels Diabetes distress (T1-DDS)

* This study was the only non-randomized control trial. PAID: Problem Areas In Diabetes scale; CIDS: Confidence in Diabetes Self-care scale; CES-D: Centre for Epidemiological Studies scale for Depression; DQoLY: Diabetes Quality of Life Measure for Youths; WBK: Wellbeing Questionnaire; PMDQ: Personal Models of Diabetes Scale; Swe-PAID 20: Swedish version of the 20-item Problem Areas In Diabetes Scale; W-BQ12: WellBeing Questionnaire; HAD: Hospital Anxiety and Depression scale; BDI: Beck Depression Inventory; WED: Wellbeing Enquiry for Diabetics; SAS: Self-Rating Anxiety Scale; SED: Self-Efficacy for Diabetes; DSQ: Diabetes Stress Questionnaire For Youths; CIS: Checklist Individual Strength, subscale fatigue severity; SIP-8: sickness impact profile-8.

## Data Availability

https://repositorio.uloyola.es/handle/20.500.12412/2365.

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
