# Peer review of "Psychotherapeutic Interventions to Improve Psychological Adjustment in Type 1 Diabetes: A Systematic Review"

_ijerph, 2021, doi:10.3390/ijerph182010940_

Round 1
Reviewer 1 Report
The main goal of this study was to systematically review the available literature on psychological treatments with a specific focus on the management of an emotional component and their impact on both glycemic control and psychological adjustment. A secondary objective was the evaluation of the differences found in adults and children-adolescents, regarding the impact of the intervention in metabolic control.
Strengths of the manuscript: clearly defined and methodologically based problem of the research. I agree, that international clinical practice guidelines highlight the importance of improving the psychological and mental health care of patients with Type 1 diabetes mellitus (T1DM). Assessment of the psychological and social status should be included as part of the comprehensive approach to people with diabetes. A systematic review is good methods for this study.
Weak parts of the manuscript: Small sample sizes - eight articles met inclusion criteria.
Methods: A systematic review was carried out to determine whether psychological treatments with a specific focus on emotional management have an impact on glycemic control and variables related to psychological adjustment. PRISMA guidelines for reporting systematic reviews were followed and the protocol was registered in PROSPERO on April 28, 2020 (registration No.: CRD42020159017) with the last update on February 23, 2021. Comprehensive literature searches of PubMed Medline, Psycinfo, Cochrane Database, Web of Science, and Open Grey Repository databases were conducted, from inception to November 2019 and were last updated in December 2020. Databases were searched separately by two reviewers (DMR and DN). The search strategy incorporated combinations of three different concepts: (a) interventions; (b) type 1 diabetes; and (c) psychological wellbeing. Search was piloted in PubMed and then adapted to run across the other databases. For the identification of additional articles, reference lists of the included studies as well as recent reviews in the field were checked. Finally, eight articles met inclusion criteria.
Results: Results showed that the management of emotions were effective in improving the psychological adjustment of patients with T1DM when carried out by psychologists. However, the evidence regarding the improvement of glycemic control was not entirely clear. When comparing adolescent and adult populations, results yielded slightly better results in adolescents.
Conclusions: More rigorous studies are needed to establish what emotional interventions might increase glycemic control in this population.
The article written in an appropriate way. The results interpreted appropriately. Article is suitable for publishing.
Author Response
We want to thank Reviewer 1 for the endorsement for publish the study
Reviewer 2 Report
This review describes the effects of psychological interventions given by psychologists to diabetes patients with respect to psychological outcomes and glycemic control. The topic of this manuscript is clinically relevant, it is well-written and wise things are said. Nevertheless, I see some major objections:
- No information is presented with regard to the magnitude of the effects, let alone on their clinical relevance. Though the authors may be correct in saying that a meta-analysis was not appropriate, there is no objection against presenting effect sizes.
- The authors propose two assumptions: (a) Interventions offered by a psychologist do probably better than offered by medical personnel, and (b) Interventions including an emotional component do probably better than interventions without it. It is a mystery why the authors did not collect information with respect to both contrasts. They limited themselves by studying interventions offered by a psychologist and including an emotional component. Now we do not know whether their assumptions are correct. That is a pity, because we do not know whether care providers should be advised to also apply these limitations.
At least, the authors could present an overview of what has been found in studies in other disease areas.
- In the discussion, the authors state: "When a patient finally experiences diabetes distress (DD), they are more likely to reduce their self-care". That is an important topic. Why did the authors not collect information with respect to the effects on self-care, somatic functioning, daily functioning and/or number of symptoms?
Minor remarks:
- R. 57-59, "Furthermore, diabetes management must consider the general state of mental health, as adding a mental health problem to the DD, could aggravate the patient’s condition". This is an unclear sentence. First, what is "the general state of mental health"? Second, as DD is a mental health problem, what does it mean that you add a mental health problem to a mental health problem?
- R 69-72: It is said that two meta-analyses found an effect on glycemic control, whereas a third one did not report such an effect. Could the authors explain this discrepancy?
- In the next sentence it is said: "On the contrary, there is research that supports that, the intensity of psychological interventions and motivational interviews are related to a reduction in DD and HbA1c". That is confusing: First, "yes, an effect was found", then "no, not in another meta-analysis", and now "yes, research has found an effect". What is meant by "the intensity of psychological studies"? Its length, its frequency of sessions, or its number of components?
- R. 82-83: The authors mention four limitations of previous reviews. One of them is: "They have included population with psychological symptoms such as subclinical depression". Why is that a problem? In my view, diabetes patients with depression should be the first to be considered for psychological treatment. Moreover, this limitation is not mentioned as an exclusion criterion for their own review.
- Of the "full-text articles assessed", seven could not be found. How can one assess a full-text article that has not been found?
- Figure 2 is incomprehensible. For three aspects of quality, there are six bars. Why is that? The heading says "As percentage". As percentage of what? What does the addition intention to treat" means at this place? Why do the bars end at 90% instead of 100%?
- Table 2 should be simplified. Even diabetes specialists will not read all the details about inclusion and exclusion criteria, let alone the great majority of readers who are not involved in diabetes care. Think what this last group should know. Maybe this column could be better deleted. Also, please do not mention statistical details, but mention only in which outcome variables an effect yes or no was found.
- R. 268-270, "The results of these studies [the studies included in the present review] showed that the ability to use emotional information may be important to understand the management of the disease in adults and adolescents with T1DM". In my view, it is clearer to say: " … may be important for diabetes patients to better manage their disease".
- R. 279-280, "Furthermore, to maximize the value of these interventions, it would be important to include them in the routine care". Some sentences give the impression that the authors believe that a complex formulation is more scientific than a simple one. Why not a formulation such as: "Given the efficacy of the interventions, it would be wise to include them in the routine care".
- R. 281-292. In this paragraph, the authors try to analyze which elements of an intervention contributed to an effect on glycemic control. However, they only mention the elements of successful interventions and forgot to mention whether these elements were absent in unsuccessful interventions.
- In the same paragraph the authors mention biofeedback as an important element of an intervention. They concluded that an emotional element was decisive. However, biofeedback is not an emotional element.
- R. 293-307. The same line of reasoning here is similar to the one in the previous paragraph. Here, the contrast is between children and adolescents versus adults. Only the effects for children and adults are mentioned. So, one cannot conclude " … that the impact of interventions with emotional management is more significant in adolescents than in adults". I would like to see a simple table with the three contrasts in rows and success or no success in columns.
- In the same paragraph, the contrast between short-term and long-term effects is mentioned, but nowhere is mentioned what short-term and long-term is. This could also be placed on the table.
Author Response
R. No information is presentedwith regard to the magnitude of the effects, let alone on their clinical relevance. Though the authors may be correct in saying that a meta-analysis was not appropriate, there is no objection against presenting effect sizes. The authors propose two assumptions: (a) Interventions offered by a psychologist do probably better than offered by medical personnel, and (b) Interventions including an emotional component do probably better than interventions without it. It is a mystery why the authors did not collect information with respect to both contrasts. They limited themselves by studying interventions offered by a psychologist and including an emotional component. Now we do not know whether their assumptions are correct. That is a pity, because we do not know whether care providers should be advised to also apply these limitations.
Thank you for your comment. The main objective of the present systematic review was to determine whether psychological treatments, carried out by psychologists, would have an impact on glycemic control and other variables related to psychological adjustment. According to the authors, it is an important issue to address psychological interventions carried out by the psychologist as the main specialist in the multidisciplinary treatment of diabetes. In addition, the authors wanted to describe the results of the interventions that include an emotional component in the diabetes management, not try to compare emotional components against other interventions.
R. At least, the authors could present an overview of what has been found in studies in other disease areas.
Thank you for your suggestion. Authors agree on the importance that emotional components may have in other areas of disease, but it is beyond the scope of this systematic review that focuses specifically on diabetes.
R. In the discussion, the authors state: "When a patient finally experiences diabetes distress (DD), they are more likely to reduce their self-care". That is an important topic.Why did the authors not collect information with respect to the effects on self-care, somatic functioning, daily functioning and/or number of symptoms?
Thank you for your comment. As Reviewer 2 states, there are other variables such as self-care or somatic functioning that are relevant for the management of diabetes. However, the present systematic review focuses on the relationship between the management of emotional components and the impact on the psychological adjustment (such as for example distress or quality of life). The authors agree with Reviewer 2 on the importance of the topic but in their opinion, it is beyond the scope of the review.
Minor remarks:
R. 57-59, "Furthermore, diabetes management must consider the general state of mental health, as adding a mental health problem to the DD, could aggravate the patient’s condition". This is an unclear sentence. First, what is "the general state of mental health"? Second, as DD is a mental health problem, what does it mean that you add a mental health problem to a mental health problem?
Thank you for the comment. The full explanation has been modified to be more understandable (lines 57-62)
“Assessment of the psychological and social status should be included as part of the comprehensive approach to people with diabetes. Systematizing the assessment of the psychological state will help mitigate the psychological impact of diabetes [13]. In fact, clinical practice guidelines [14] highlight the importance of improving care for the psychological and mental health problems of patients with T1DM.”
R.69-72: It is said that two meta-analyses found an effect on glycemic control, whereas a third one did not report such an effect. Could the authors explain this discrepancy?
Thank you for your comment. The discrepancy has been explained in more detail (lines 78-81)
The discrepancy between results could be explained as a consequence of the limited number of studies identified [20] or because the other metanalyses were either focused on mindfulness-based interventions [19] or included only adolescent population [18].
R. In the next sentence it is said: "On the contrary, there is research that supports that, the intensity of psychological interventions and motivational interviews are related to a reduction in DD and HbA1c". That is confusing: First, "yes, an effect was found", then "no, not in another meta-analysis", and now "yes, research has found an effect". What is meant by "the intensity of psychological studies"? Its length, its frequency of sessions, or its number of components?
“On the contrary” has been changed for “In this sense”. In addition…
Authors specified what “intensity of psychological interventions” means (lines 82-83):
“the intensity of psychological interventions (determined by the number and duration of sessions)”
R. 82-83: The authors mention four limitations of previous reviews. One of them is: "They have included population with psychological symptoms such as subclinical depression". Why is that a problem? In my view, diabetes patients with depression should be the first to be considered for psychological treatment. Moreover, this limitation is not mentioned as an exclusion criterion for their own review.
Thank you for your comment. Authors identified different limitations of previous reviews in order to define their inclusion criteria. For the present systematic review, participants with any concurrent psychiatric disease were excluded.
R. Of the "full-text articles assessed", seven could not be found. How can one assess a full-text article that has not been found?
Thank you for your comment. The previous sentence was mistaken. It has been changed (line 160)
“Of the 79 articles selected for full text reviewing, 72 were excluded for the following main reasons”.
R. Figure 2 is incomprehensible. For three aspects of quality, there are six bars. Why is that? The headingsays "As percentage". As percentage of what? What does the addition intention to treat" means at this place? Why do the bars end at 90% instead of 100%?
There are six bars because there are six domains assessed with Rob2 Tool. There was a mistake in copy pasting the image and only three columns were detailed. The authors have addressed this issue.
Intention to treat references to how the analyses were carried out in the studies. When analyzing the quality of the studies with Rob2Tool, this is one of the questions that should be responded.
R. Table 2 should be simplified. Even diabetes specialists will not read all the details about inclusion and exclusion criteria, let alone the great majority of readers who are not involved in diabetes care. Think what this last group should know. Maybe this column could be better deleted. Also, please do not mention statistical details, but mention only in which outcome variables an effect yes or no was found.
The authors have modified the format of Table 2.
R. 268-270, "The results of these studies [the studies included in the present review] showed that the ability to use emotional information may be important to understand the management of the disease in adults and adolescents with T1DM". In my view, it is clearer to say: " … may be important for diabetes patients to better manage their disease".
Thank you for your comment. The sentence has been restructured (lines 274-275):
“The results of the studies included in the present review showed that the ability to use emotional information may be important for patients with T1DM to better manage their disease.”
R. 279-280, "Furthermore, to maximize the value of these interventions, it would be important to include them in the routine care". Some sentences give the impression that the authors believe that a complex formulation is more scientific than a simple one. Why not a formulation such as: "Given the efficacy of the interventions, it would be wise to include them in the routine care".
Thank you for your comment. The sentence has been modified (lines 284-285):
“Furthermore, given the efficacy of the interventions it would be important to include them in routine care.”
R. 281-292. In this paragraph, the authors try to analyze which elements of an intervention contributed to an effect on glycemic control. However, they only mention the elements of successful interventions and forgot to mention whether these elements were absent in unsuccessful interventions.In the same paragraph the authors mention biofeedback as an important element of an intervention. They concluded that an emotional element was decisive. However, biofeedback is not an emotional element.
Thank you for your comment. The authors have changed the statement to improve the paragraph (lines 292-299)
“In this sense, the studies that did not find reductions in HbA1c were delivered either only in group format or individually or family format. Second, self-care behaviors were central and explicit targets in the interventions, which can help maintain adequate HbA1c level. Third, some of the studies have patients with elevated HbA1c at baseline which may respond better to the intervention. Psychological interventions with an emotional component might generate an improvement in HbA1c levels by reducing DD and stress levels [36, 37]. However, more studies are needed to explore this relationship.”
R. 293-307. The same line of reasoning here issimilar tothe one in the previous paragraph. Here, the contrast is between children and adolescents versus adults. Only the effects for children and adults are mentioned. So, one cannot conclude " … that the impact of interventions with emotional management is more significant in adolescents than in adults". I would like to see a simple table with the three contrasts in rows and success or no success in columns. In the same paragraph, the contrast between short-term and long-term effects is mentioned, but nowhere is mentioned what short-term and long-term is. This could also be placed on the table.
The authors have modified the statement (lines 308-315):
“These results are similar to the studies that included adult population [28, 32, 33]. Even though studies with adolescents included in the present systematic review are fewer than those carried out with adults, it can be pointed out that the impact of interventions with emotional management could be more relevant in adolescents due to the difficulties of glycaemic control at this stage of life. However, one thing in common is the effect of the interventions over time. Since in both populations the long-term effects (12 months) are more visible and stable than in the short-term (3 months). These results are in line with those found in previous metanalyses [18, 19].”
Reviewer 3 Report
My main comments and suggestions are that authors of this paper do not mention the attachment-based strategies as the appropriate psychotherapeutic interventions in the management of T1DM. Even if it is clear that randomized controlled trials studies with attachment-based interventions in T1DM are lacking so far, my suggestion is to mention the attachment-oriented approach on some places (introduction, discussion, limitation, future directions...).
I mention below results of some studies.
An interaction between attachment and communication quality was significantly associated with glycosylated hemoglobin (Hb A(1c)) levels and dismissing (avoidant) attachment in the setting of poor patient-provider communication is associated with poorer treatment adherence in patients with diabetes
- Ciechanowski, P. S., Katon, W. J., Russo, J. E., & Walker, E. A. (2001). The patient-provider relationship: attachment theory and adherence to treatment in diabetes. American Journal of Psychiatry, 158(1), 29-35.
Attachment theory’s central hypothesis is that a secure relationship with a caregiver in the early life of a child is essential to normal emotional and relational development. Authors of recent papers extend the ideas of attachment, into the psychological adaptation processes for young people at the time of diagnosis of diabetes1 with emphasis on the function of the parent/caregiver in mentalising the experience of the child.
- Garrett, C. J., Ismail, K., & Fonagy, P. (2021). Understanding developmental psychopathology in Type 1 diabetes through attachment, mentalisation and diabetes distress. Clinical Child Psychology and Psychiatry, 1359104521994640) function of the parent/caregiver in mentalising the experience of the child.
- Oldham-Cooper, R., Semple, C., & Wilkinson, L. L. (2021). Reconsidering a role for attachment in eating disorder management in the context of paediatric diabetes. Clinical Child Psychology and Psychiatry, 1359104520986215.
- Costa-Cordella, S., Luyten, P., Cohen, D., Mena, F., & Fonagy, P. (2021). Mentalizing in mothers and children with type 1 diabetes. Development and psychopathology, 33(1), 216-225
According to one study was maternal perceptions of more secure attachment associated with better glycemic control.
-Rosenberg, T., & Shields, C. G. (2009). The role of parent–adolescent attachment in the glycemic control of adolescents with type 1 diabetes: A pilot study. Families, Systems, & Health, 27(3), 237.) was maternal perceptions of more secure attachment associated with better glycemic control.
Another recent study showed that the presence of an insecure attachment, especially to the mother, worsens the psychological adaptation of T1DM children. T1DM children with insecure attachment to mother scored significantly higher in anxious/depressed, withdrawn/depressed, attention problems, and rule-breaking behavior.
-Bizzi, F., Della Vedova, A. M., Prandi, E., Cavanna, D., & Manfredi, P. (2021). Attachment representations to parents and emotional-behavioral problems: A comparison between children with type 1 diabetes mellitus and healthy children in middle childhood. Clinical Child Psychology and Psychiatry, 26(2), 393-405
Higher anxious and avoidant attachment is associated with higher diabetes distress. Anxious attachment is associated with higher daily stressors and lower self-care.
- Kelly, C. S., Berg, C. A., & Helgeson, V. S. (2020). Adult attachment insecurity and associations with diabetes distress, daily stressful events and self-management in type 1 diabetes. Journal of behavioral medicine, 43(5), 695-706.
When authors emphasize the effectivity of psychological interventions with an emotional component in patients with T1DM they should mention the alexithymia, as alexithymia and attachment could affect selfcare and blood glucose level in adolescents with type 1 diabetes
-Shayeghian, Z., Moeineslam, M., Hajati, E., Karimi, M., Amirshekari, G., & Amiri, P. (2020). The relation of alexithymia and attachment with type 1 diabetes management in adolescents: a gender-specific analysis. BMC psychology, 8(1), 1-9.
In the Materials and Methods section is some redundant probably copied paragraph (lines 102-108), please delete.
Author Response
R. My main comments and suggestions are that authors of this paper do not mention the attachment-based strategies as the appropriate psychotherapeutic interventions in the management of T1DM. Even if it is clear that randomized controlled trials studies with attachment-based interventions in T1DM are lacking so far, my suggestion is to mention the attachment-oriented approach on some places (introduction, discussion, limitation, future directions...).
I mention below results of some studies.
An interaction between attachment and communication quality was significantly associated with glycosylated hemoglobin (Hb A(1c)) levels and dismissing (avoidant) attachment in the setting of poor patient-provider communication is associated with poorer treatment adherence in patients with diabetes
- Ciechanowski, P. S., Katon, W. J., Russo, J. E., & Walker, E. A. (2001). The patient-provider relationship: attachment theory and adherence to treatment in diabetes. American Journal of Psychiatry, 158(1), 29-35.
Attachment theory’s central hypothesis is that a secure relationship with a caregiver in the early life of a child is essential to normal emotional and relational development. Authors of recent papers extend the ideas of attachment, into the psychological adaptation processes for young people at the time of diagnosis of diabetes1 with emphasis on the function of the parent/caregiver in mentalising the experience of the child.
- Garrett, C. J., Ismail, K., & Fonagy, P. (2021). Understanding developmental psychopathology in Type 1 diabetes through attachment, mentalisation and diabetes distress. Clinical Child Psychology and Psychiatry, 1359104521994640) function of the parent/caregiver in mentalising the experience of the child.
- Oldham-Cooper, R., Semple, C., & Wilkinson, L. L. (2021). Reconsidering a role for attachment in eating disorder management in the context of paediatric diabetes. Clinical Child Psychology and Psychiatry, 1359104520986215.
- Costa-Cordella, S., Luyten, P., Cohen, D., Mena, F., & Fonagy, P. (2021). Mentalizing in mothers and children with type 1 diabetes. Development and psychopathology, 33(1), 216-225
According to one study was maternal perceptions of more secure attachment associated with better glycemic control.
-Rosenberg, T., & Shields, C. G. (2009). The role of parent–adolescent attachment in the glycemic control of adolescents with type 1 diabetes: A pilot study. Families, Systems, & Health, 27(3), 237.) was maternal perceptions of more secure attachment associated with better glycemic control.
Another recent study showed that the presence of an insecure attachment, especially to the mother, worsens the psychological adaptation of T1DM children. T1DM children with insecure attachment to mother scored significantly higher in anxious/depressed, withdrawn/depressed, attention problems, and rule-breaking behavior.
-Bizzi, F., Della Vedova, A. M., Prandi, E., Cavanna, D., & Manfredi, P. (2021). Attachment representations to parents and emotional-behavioral problems: A comparison between children with type 1 diabetes mellitus and healthy children in middle childhood. Clinical Child Psychology and Psychiatry, 26(2), 393-405
Higher anxious and avoidant attachment is associated with higher diabetes distress. Anxious attachment is associated with higher daily stressors and lower self-care.
- Kelly, C. S., Berg, C. A., & Helgeson, V. S. (2020). Adult attachment insecurity and associations with diabetes distress, daily stressful events and self-management in type 1 diabetes. Journal of behavioral medicine, 43(5), 695-706.
When authors emphasize the effectivity of psychological interventions with an emotional component in patients with T1DM they should mention the alexithymia, as alexithymia and attachment could affect selfcare and blood glucose level in adolescents with type 1 diabetes
-Shayeghian, Z., Moeineslam, M., Hajati, E., Karimi, M., Amirshekari, G., & Amiri, P. (2020). The relation of alexithymia and attachment with type 1 diabetes management in adolescents: a gender-specific analysis. BMC psychology, 8(1), 1-9.
Thank you for your comment and references suggestions, the authors have modified the manuscript accordingly
Introduction (lines 67-70)
“Some studies emphasize the association between attachment orientation and adherence to glucose monitoring. Attachment theory supports that a secure relationship with the caregiver in early life will lead to normal emotional development and better adaptation to diabetes [16]”
Discussion (lines 266-269)
“Attachment theory supports the fact that patients internalize early experiences with caregivers that will determine whether others can be trusted, this influences their participation in the patient-provider relationship and ultimately, adherence to treatment [35]”.
Discussion (future lines) (lines 335-336):
“Future studies in this regard may be oriented to explore the psychologist-patient relationship within the framework of the attachment theory”.
The following references have been included:
16. Oldham-Cooper R, Semple C, Wilkinson LL. Reconsidering a role for attachment in eating disorder management in the context ofpaediatric diabetes. Clin Child Psychol Psychiatry. 2021, 26, 669-681. https://doi.org/10.1177/1359104520986215.
35. Ciechanowski, P. S., Katon, W. J., Russo, J. E., & Walker, E. A. The patient-provider relationship: attachment theory and adherence to treatment in diabetes. American Journal of Psychiatry, 2001,158, 29-35. https://doi.org/10.1176/appi.ajp.158.1.29
R. In the Materials and Methods section is some redundant probably copied paragraph (lines 102-108), please delete.
Thank you for your comment. The authors have deleted these sentences.
Round 2
Reviewer 2 Report
I had three main problems with the previous version, which I reproduce below. None of these problems were addressed by the authors. The authors only responded by saying something like "We have a simple question, and we do not want more than answering this simple question".
Their approach is comparable to writing a review of the effects of a special medicine A given by a nurse. There is information available about the effects of medicine B and C, and of medicine A given by a medical doctor. However, the reviewers prefer to ignore this information, and - yes - they presented evidence that medicine A given by a nurse is efficacious. However, the reader of this review does not know whether the effect is clinically relevant, nor does he know whether it is advisable to prescribe medicine A, B, or C, nor does he know whether medicine A should be given by a nurse or a medical doctor. This is a short-sighted, not very scientific and not clinically relevant approach
- No information is presented with regard to the magnitude of the effects, let alone on their clinical relevance. Though the authors may be correct in saying that a meta-analysis was not appropriate, there is no objection against presenting effect sizes.
- The authors propose two assumptions: (a) Interventions offered by a psychologist do probably better than offered by medical personnel, and (b) Interventions including an emotional component do probably better than interventions without it. It is a mystery why the authors did not collect information with respect to both contrasts. They limited themselves by studying interventions offered by a psychologist and including an emotional component. Now we do not know whether their assumptions are correct. That is a pity, because we do not know whether care providers should be advised to also apply these limitations. At least, the authors could present an overview of what has been found in studies in other disease areas.
- In the discussion, the authors state: "When a patient finally experiences diabetes distress (DD), they are more likely to reduce their self-care". That is an important topic. Why did the authors not collect information with respect to the effects on self-care, somatic functioning, daily functioning and/or number of symptoms?
Author Response
First of all we would like to thank the reviewer for investing time and effort to make our work more valuable. In addition, we have upload the manuscript in the website. Specific responses to the comments are set out below:
1. No information is presented with regard to the magnitude of the effects, let alone on their clinical relevance. Though the authors may be correct in saying that a meta-analysis was not appropriate, there is no objection against presenting effect sizes.
We deleted the column with statistics from table 2 suggested by this same reviewer. We included that information at first, because almost none of the studies reported specific magnitude effects.
2. The authors propose two assumptions: (a) Interventions offered by a psychologist do probably better than offered by medical personnel, and (b) Interventions including an emotional component do probably better than interventions without it. It is a mystery why the authors did not collect information with respect to both contrasts. They limited themselves by studying interventions offered by a psychologist and including an emotional component. Now we do not know whether their assumptions are correct. That is a pity, because we do not know whether care providers should be advised to also apply these limitations. At least, the authors could present an overview of what has been found in studies in other disease areas.
In our previous answer we tried to explain that the scope of our study was only to assess if psychological interventions with an emotional component provided by psychologists were effective, not to test whether they were more effective than with other professionals and components. We agree that this is a drawback from our study and as such we have reflected it in our manuscript.
(Line 321) Second, this study has evaluated the effect of psychological interventions with an emotional component delivered by psychologists but its comparative effect with respect to other interventions or health professionals has not been evaluated.
Regarding the inclusion of studies addressing this issue in other diseases we have included a brief comment on the subject and new references.
(Line 267) Recent studies on chronic diseases indicate that physicians, nurses and other health professionals often lack the training and skills that psychologist have, to deliver behavior-based treatments, and also highlight that interprofessional teams yield the most positive outcomes [36-38].
36. Ockene JK, Ashe K, Peterson KS, Fitzgibbon M, Buscemi J, Dulin A. Society of Behavioral Medicine Call to Action: Include obesity/overweight management education in health professional curricula and provide coverage for behavior-based treatments of obesity/overweight most commonly provided by psychologists, dieticians, counselors, and other health care professionals and include such providers on all multidisciplinary teams treating patients who have overweight or obesity. Transl Behav Med. 2021, 11(2):653-655. https://doi.org/10.1093/tbm/ibaa030
37. Rich K, Murray K, Smith H, Jelbart N. Interprofessional practice in health: A qualitative study in psychologists, exercise physiologists, and dietitians. J Interprof Care. 2021, 35(5):682-690. https://doi.org/10.1080/13561820.2020.1803226
38. García-Llana H, Barbero J, Olea T, Jiménez C, Del Peso G, Miguel JL, Sánchez R, Celadilla O, Trocoli F, Argüello MT, Selgas R. Incorporación de un psicólogo en un servicio de nefrología: criterios y proceso [Incorporation of a psychologist into a nephrology service: criteria and process]. Nefrologia. 2010, 30(3):297-303. doi: 10.3265/Nefrologia.pre2010.Apr.10407.
3. In the discussion, the authors state: "When a patient finally experiences diabetes distress (DD), they are more likely to reduce their self-care". That is an important topic. Why did the authors not collect information with respect to the effects on self-care, somatic functioning, daily functioning and/or number of symptoms?
Effects of the training on specific aspects of self-care like somatic or daily functioning were beyond the scope of the review which was focused on psychological adjustment. This topic is large and complex enough for another review.
Reviewer 3 Report
Dear authors,
I agree with current version of your manuscript.